# High-Fat Diet with Normal Caloric Intake Elevates TMA and TMAO Production and Reduces Microbial Diversity in Rats

**DOI:** 10.3390/nu17132230

**Published:** 2025-07-05

**Authors:** Mateusz Szudzik, Mikołaj Zajdel, Emilia Samborowska, Karol Perlejewski, Marek Radkowski, Marcin Ufnal

**Affiliations:** 1Laboratory of Centre for Preclinical Research, Department of Experimental Physiology and Pathophysiology, Medical University of Warsaw, 02-091 Warsaw, Poland; s082807@student.wum.edu.pl; 2Mass Spectrometry Laboratory, Institute of Biochemistry and Biophysics, Polish Academy of Sciences, 02-106 Warsaw, Poland; emi.sambor@gmail.com; 3Department of Immunopathology of Infectious and Parasitic Diseases, Medical University of Warsaw, 02-106 Warsaw, Poland; karol.perlejewski@wum.edu.pl (K.P.); marek.radkowski@wum.edu.pl (M.R.)

**Keywords:** high-fat diet, trimethylamine, trimethylamine *N*-oxide, microbiome

## Abstract

**Background/Objectives:** Trimethylamine (TMA), produced by gut microbiota, and its derivative trimethylamine N-oxide (TMAO) are both associated with cardiometabolic diseases. While the effects of high-fat diets (HFDs) and high-disaccharide diets (HDDs) on gut microbiota in the context of obesity have been well studied, their impact on TMA/TMAO production, particularly alongside physiological caloric intake, remains obscure. This study investigates how standard HFDs and HDDs alongside physiological caloric intake influence gut microbiota composition and TMA/TMAO production in rats. **Methods:** Sprague Dawley rats were fed one of three diets a standard diet, an HFD, or an HDD for 12 weeks, with chow availability adjusted by age to maintain physiological caloric intake. Gut bacterial diversity was analyzed using 16S rRNA gene sequencing, and metabolites were quantified via High-Performance Liquid Chromatography-Mass Spectrometry (HPLC-MS) in urine and plasma. **Results:** The HFD group had significantly higher urinary levels of TMA and TMAO compared to the control and HDD groups. Gut bacterial diversity in the HFD group was markedly reduced, displaying the lowest species richness and phylogenetic diversity among all the groups. Notably, Pasteurellaceae (within the order Pasteurellales) and S24-7 (within the order Bacteroidales) were positively correlated with TMAO levels. The demonstrated HDD group increased microbial diversity compared to both the control and HFD groups. **Conclusions:** A high-fat diet during controlled and physiological caloric intake increases TMA/TMAO production and reduces gut microbial diversity. This underscores the role of diet composition, beyond caloric excess, in shaping gut microbiota and the related cardiometabolic biomarkers.

## 1. Introduction

Trimethylamine (TMA) and its oxidized derivative, trimethylamine *N*-oxide (TMAO), are derived from the gut microbiota metabolism of dietary carnitine and choline [1]. These metabolites have garnered significant attention due to their association with cardiometabolic dysfunction, where they are recognized as both biomarkers and potential mediators of cardiovascular disease [2,3,4].

Recent research suggests that specific bacterial taxa including Clostridium, Escherichia, and Desulfovibrio are involved in TMA production; yet, the extent to which these findings can be generalized across various dietary and metabolic contexts remains unclear [5,6].

Dietary composition is known to be a significant determinant of TMA and TMAO production, with evidence showing that high-fat and high-choline diets promote the abundance of TMA-producing bacteria and elevate plasma TMAO [3,7]. However, studies to date have employed ad libitum feeding models, which lead to obesity complicating the interpretation of findings. In contrast, the aim of this study was to investigate the effects of HFD and HDD with physiological caloric intake on TMA and TMAO production, a model free from the confounding influence of obesity.

## 2. Materials and Methods

### 2.1. Animals and Experimental Design

The experiments adhered to Directive 2010/63 EU, which focuses on safeguarding animals used for scientific purposes. These experiments were also approved by the Local Bioethical Committee (no: WAW2/107/2021). The rats were housed in groups of 4 animals in polypropylene cages with environmental enrichment, following a 12 h light and 12 h dark cycle. The temperature was maintained at 22–23 °C, with a humidity level of 45–55%.

The samples for the present study were obtained from our previous study [8] on male and female Sprague Dawley rats (*n* = 44) divided into three groups. The control group (*n* = 16) received a standard laboratory diet, while the high-disaccharides diet group (*n* = 14) (HDD) and the high-fat diet (HFD) group (*n* = 14) received diets rich in fats during the 3 months. The animals were allocated into groups based on their initial body weight, ensuring that average body weight did not differ significantly between the groups at the beginning of the experiment.

The composition and nutritional value of the diets are shown in Table 1. Throughout the experiment, all animals, regardless of the supplementation procedure, received the same caloric intake daily. All experimental groups received an equivalent caloric intake, adjusted according to body weight gain as the rats matured. The rats were weighed weekly, and their energy needs were calculated based on their current weights using guidelines from the Nutrient Requirements of Laboratory [9]. Each day they were provided with a specific amount of feed (in grams) corresponding to their calculated caloric requirement. The rats were housed in cages with a fixed amount of food tailored to their body weight. Any uneaten food was collected and weighed daily, with the leftover amount converted into caloric values. The actual energy intake for each rat over the course of the experiment was determined by subtracting the caloric value of the uneaten food from the total calories initially provided. Appendix A provides details of the total calorie intakes. Appendix A presents the initial and final body weight of the rats.

The rats were initially anesthetized via an intraperitoneal (IP) injection of a ketamine-xylazine mixture (70 mg/kg ketamine and 10 mg/kg xylazine). Once fully anesthetized, blood was collected directly from the heart using a syringe. Blood samples were collected using chilled EDTA tubes and centrifuged at 5000 rpm for 5 min at 4 °C. The resulting plasma was transferred into Eppendorf tubes and stored at −20 °C. Following blood collection, the rats were euthanized by dislocation, and stool samples (0.5 mL) were obtained from the midsection of the colon. The animals were fasted for 6 h before sacrifice.

The rats were maintained for 2 days in metabolic cages to evaluate their 24 h fluid and energy balance prior to and after the main section of this study. Body mass, energy and water intake, and urine output were measured at the start and at the end of the experiment [8].

### 2.2. 16 rRNA Library Preparation and Sequencing

DNA was isolated from 180 to 220 mg of stool with the use of a Nucleospin DNA Stool Kit (Macherey-Nagel; Düren, Germany), following the manufacturer’s protocol. Final elution was performed in 100 μL of water. DNA was quantified using a Qubit High Sensitivity Kit (ThermoFisher, Waltham, MA, USA). The V3 and V4 regions of the 16S rRNA gene were amplified (550 bp product) with the use of 12 ng of extracted DNA, a set of V3–V4 primers [10], and 0.5 U of polymerase from a KAPA HiFi HotStart ReadyMix PCR Kit (Roche Molecular Diagnostics, Basel, Switzerland). The amplified products were purified using a 0.8 ratio of AMPure XP beads (Beckman Coulter Life Sciences, Indianapolis, IN, USA) and their size was evaluated with a Bioanalyzer and DNA 1000 kit (Agilent Technologies, Santa Clara, CA, USA). Sequencing libraries were dual-indexed using the Nextera XT Index kit (Illumina, San Diego, CA, USA) and sequenced (300 nt, paired-end reads) using a MiSeq Reagent Kit v3 (600-cycle) on the Illumina MiSeq platform.

### 2.3. Bioinformatic and Statistics

FastQC software (v0.11.7) was used to evaluate the quality of the sequencing reads [11], which were further trimmed by trimmomatic [12] and filtered based on their size by BBTools [13]. The remaining reads were subjected to an analysis with QIIME (version 1.9.1); [14]. In short, the forward and reverse reads were merged using fastq-join command [14]. Operational taxonomic units (OTUs) were determined by an open-reference OTU picking process and the Greengenes database (threshold value of 97% sequence similarity). Unmapped sequences were clustered de novo and aligned once again. Chimera detection was performed with the use of ChimeraSlayer [15]. Normalization was achieved using metagenomeSeq’s CSS (cumulative sum scaling) transformation [16].

A Linear discriminant analysis (LDA) effect size (LEfSe) analysis was performed to detect differentially abundant bacterial taxa [17]. The identified taxa with an LDA score higher than two were considered to be enriched in the respective group as compared to the other groups.

A statistical analysis was performed based on alpha- and beta-diversity metrics. Principal Coordinates Analysis (PCoA) plots (unweighted and weighted UniFrac data) for beta diversity were generated using QIIME output data in PhyloToAST [18]. Abundance plots were generated using the phyloseq package in R (v1.38.0) [19]. The Mann–Whitney U test was used for alpha diversity statistics, whereas a taxonomical comparison between the different groups was performed using a nonparametric *t*-test. Additionally, a categorical variable analysis of similarities (ANOSIM) was performed.

### 2.4. Biochemicals and Metabolites Panel Analysis

Plasma and urine concentrations of bacterial metabolites were measured using Waters Acquity Ultra Performance Liquid Chromatograph (Waters Corporation, Milford, MA, USA) coupled with a Waters TQ-S triple-quadrupole mass spectrometer (Waters, Manchester, UK). The mass spectrometer was operated in the multiple reaction monitoring (MRM)-positive electrospray ionization (ESI) mode, as previously described [20].

### 2.5. Statistical Analysis

The Shapiro–Wilk test was used to test the normality of the distribution. The results were statistically analyzed with ANOVA followed by Tukey’s post hoc test for normally distributed data or a Mann–Whitney test for nonnormal distributed data using GraphPad Prism version 8.0.1 for Windows, (GraphPad Software, La Jolla, CA, USA). *p* ≤ 0.05 or lower, was considered significant.

## 3. Results

### 3.1. Gut Microbiota Composition in Rats Receiving Normal Chow vs. An HFD for 12 Weeks

#### 3.1.1. Start of the Experiment

There were no significant differences in the composition of gut microbiota between the given groups at the start of the experiment (Figure 1a).

#### 3.1.2. Dietary Interventions Altered the Microbiomes

Differences in gut bacteria composition between the start and the end of the experiment were found in all the presented groups. A beta diversity analysis (weighted and unweighted) showed significant differences (*p* < 0.001) between the groups (Figure 1a).

#### 3.1.3. The HDD Enhanced Microbial Diversity, Whereas the HFD Resulted in Its Decline

The beta diversity analysis (weighted and unweighted) showed significant differences (*p* < 0.001) between the groups (Figure 1a). A comparison of alpha diversity (within-sample microbial diversity) showed that the HDD group has a higher average number of species and higher phylogenetic diversity than the controls and HFD, while the HFD group has a significantly lower average number of species and phylogenetic diversity compared to the controls (Figure 1a). Significant differences in the Shannon parameters and observed OTUs between the HFD group and the controls were found (Figure 1b).

Using the pair abundancy statistic (FDR P), the abundance of bacteria was unchanged while the unpaired statistic revealed significant differences in family and genus levels (Figure 2a).

#### 3.1.4. LDA Score

The enrichment or depletion of taxa across various taxonomic levels, from phylum to genus, was identified through an LEfSe analysis (Figure 2b).

The taxa that were enriched in the controls were as follows: at the phylum level, Bacteroidetes, Verrucomicrobia, and Proteobacteria; at the class level, Bacteroidia, and Deltaproteobacteria; at the order level, Bacteroidales, Verrucomicrobiales, Desulfovibrionales Actinomycetales, and Lactobacillales; at the family level, Ruminococcaceae, Verrucomicrobiaceae Desulfovibrionaceae, and Streptococcaceae Micrococcaceae; and at the genus level, Ruminococcus, Phascolarctobacterium, Rothia, Kocuria, and Clostridium.

The enriched taxa in the HFD rats were as follows: at the phylum level, Firmicutes, and Tenericutes; at the class level, Actinobacteria, Mollicutes, and unidentified TM7_3; at the order level, Lactobacillales, Turicibacterales Bifidobacteriales, and unidentified CW040; at the family level, lactobacillaceae, Lachnospiraceae, Turicibacteraceae, Bifidobacteriaceae, and unidentified F16; and at the genus level, Lactobacillus, Ruminococcus, Turicibacter, Allobaculum, Bifidobacterium, and Anaeroplasma.

The enriched taxa in the HDD rats were as follows: at the family level, Ruminococcaceae, Christensenellaceae, and Mogibacteriaceae; and at the genus level, Coprobacillus, Anaerostipes, Christensenella, and Desulfovibrio.

### 3.2. The HFD Increased Urine Metabolite Levels

There were significant differences between the groups in terms of various urine metabolite levels. The HFD rats had significantly higher levels of TMA, (*p* < 0.001 vs. control; *p* < 0.05 vs. HDD), TMAO (*p* < 0.001 vs. control; *p* < 0.05 vs. HDD), betaine (*p* < 0.001 vs. control; *p* < 0.05 vs. HDD) (Table 2), and carnitine (*p* < 0.001 vs. control; *p* < 0.001 vs. HDD).

### 3.3. Daily Urinary Excretion

The HDD rats showed significantly lower TMAO excretion than the controls (*p* < 0.01) and the HFD rats (*p* < 0.001). The HFD rats had significantly higher TMA (*p* < 0.05 vs. control; *p* < 0.01 vs. HDD), betaine (*p* < 0.01 vs. control; *p* < 0.001 vs. HDD), and carnitine (*p* < 0.05 vs. control; *p* < 0.001 vs. HDD) excretions than the control and HDD rats, respectively (Table 3).

### 3.4. HDD Increased Plasma Betaine Levels

Plasma betaine level was significantly higher (*p* < 0.01) in the HDD rats compared to the controls. No differences were found between the control and HFD rats. There were no significant differences between the groups in terms of plasma TMA and TMAO levels (Table 4).

### 3.5. Correlation Analysis Between Bacterial Taxonomic Groups (Based on LDA Scores) and the Concentrations of Various Metabolites in Urine

At the phylum level, only one statistically significant correlation was observed. Tenericutes were negatively correlated with TMA under HFD conditions (r = −0.7235, *p* = 0.0457).

At the order level, under HFD conditions, Pasteurellales showed a strong positive correlation with choline (r = 0.8105, *p* = 0.0137), GPC (r = 0.7746, *p* = 0.0420), and TMAO (r = 0.7746, *p* = 0.0420). Additionally, under HDD conditions, Bifidobacteriales demonstrated a significant positive correlation with TMA (r = 0.8122, *p* = 0.0129).

At the family level, Pasteurellaceae, a member of the Pasteurellales order, was positively correlated with GPC (r = 0.8105, *p* = 0.0273) and TMAO (r = 0.7746, *p* = 0.0420) under HFD conditions. Furthermore, the S24-7 family within the Bacteroidales order was positively correlated with TMAO (r = 0.7813, *p* = 0.0420) under HFD conditions. Under HDD conditions, Bifidobacteriaceae exhibited a strong positive correlation with TMA (r = 0.8122, *p* = 0.0257).

At the genus level, Bifidobacterium, within the Bifidobacteriaceae family, was specifically associated with TMA under HDD conditions, showing a positive correlation (r = 0.8122, *p* = 0.0484) (Table 5).

## 4. Discussion

In this study, we found that an HFD and an HDD, administered during physiological caloric intake, significantly altered both the gut bacterial composition and the urinary excretion of bacterial metabolites. Specifically, we observed considerable disparities in both the alpha and beta diversity of bacteria, as well as notable differences in the specific bacterial genera and families among the experimental groups. Notably, the high-fat diet group demonstrated the most substantial variability across all the aforementioned parameters, while displaying the lowest bacterial diversity compared to the control groups. Previous research suggests that dietary components can affect gut bacteria composition [21,22]. Both carbohydrates and fats could alter gut microbiota diversity [23,24,25]. Additionally, the LDA score analysis showed specific differences on various taxonomic levels in bacterial abundance between the groups. Our findings reinforce the body of evidence that dietary fat can influence gut microbiota composition [23,24]. Furthermore, a novel finding of our study is the demonstration that excessive caloric intake is not a prerequisite for such alterations.

A metabolomic analysis revealed significantly higher levels of TMA and TMAO, betaine, and carnitine in the urine of the HFD-fed rats relative to the HDD and control groups. Furthermore, TMA and TMAO are known to be linked with increased cardiovascular and renal disease risks, hypertension, and atherogenesis, highlighting the potential metabolic pathways through which HFD exacerbates CV risk [1,2,3,26,27]. Since the levels of those metabolites were significantly lower in the HDD group, which was comparable to the control group, it suggests that the HDD does not affect the TMA/TMAO bacterial metabolic pathways. Under HFD conditions, we observed positive correlations between the order Pasteurellales and metabolites such as choline, GPC, and TMAO, suggesting their potential involvement in TMA and TMAO formation pathways in this dietary context. Similarly, the family S24-7 showed a correlation with TMAO production.

In the HDD group, a positive correlation was identified between Bifidobacteriales and TMA levels, highlighting diet-specific microbial contributions to TMA metabolism. However, overall TMA production was higher in the HFD group than in the HDD.

It is important to note that despite significant differences in the daily urine excretions of TMA and TMAO between the control and HFD rats, plasma levels remained comparable. This suggests that despite the markedly higher production and turnover of TMA and TMAO in the HFD rats, plasma concentrations are tightly regulated, at least as long as kidney function remains normal. However, it has been observed that in chronic kidney disease, there is an accumulation of TMA and TMAO due to impaired renal elimination [28]. Thus, urinary concentrations—particularly 24 h excretion levels—should be considered more reliable indicators of TMA and TMAO production than blood levels.

### Study Limitations

Although we observed correlations between urine TMA/TMAO levels and the relative abundance of Pasteurellaceae and S24-7, it is important to note that taxonomic associations alone do not confirm functional involvement in TMA production.

While these findings emphasize the influence of diet on microbiota composition and TMA/TMAO production, there is currently no conclusive evidence in the literature that Pasteurellaceae, S24-7, Tenericutes, or Bifidobacteriales directly produce TMA from precursors such as choline or L-carnitine. The enzymes essential for this process, such as cutC, have not been confirmed in their genomes. Further metagenomic or transcriptomic analyses are needed to elucidate the potential role of these microbial groups in TMA/TMAO metabolism.

## 5. Conclusions

In conclusion, our study underscores the pivotal role of diet composition, beyond mere caloric intake, in shaping gut microbiota and their metabolite production. The findings provide evidence that a high-fat diet, even with normal caloric intake, promotes a gut environment conducive to the production of TMA, a compound implicated in cardiovascular pathology. In contrast, disaccharide-rich diets do not appear to have the same effect. These results suggest that microbiota-derived metabolites may play an important role in the cardiovascular pathology associated with high-fat diets.

## Figures and Tables

**Figure 1 nutrients-17-02230-f001:**
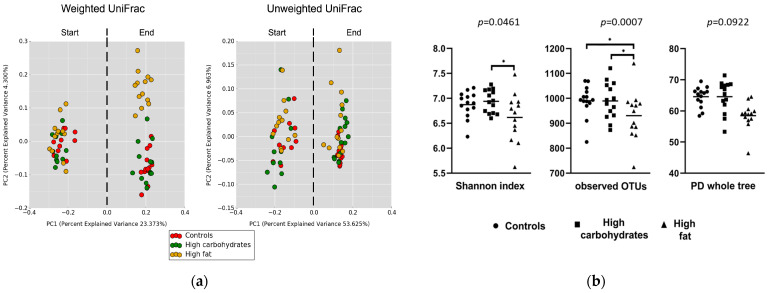
Changes in gut microbiota composition in Sprague Dawley rats maintained on a control or a high-fat diet (HFD) for 12 weeks. (**a**) Beta diversity plot showing the shift in gut microbiota using a weighted UniFrac and an unweighted UniFrac at the start and the end of experiment. Both metrics are used to compute the distances between samples and then visualize them using a Principal Coordinates Analysis (PCoA). Unweighted UniFrac measures beta diversity based on the presence or absence of taxa, focusing on community membership and giving weight to rare organisms. In contrast, weighted UniFrac incorporates both presence and relative abundance, highlighting the differences in community structure and emphasizing dominant taxa. (**b**) Alfa diversity metrics showing community richness in the studied groups: Shannon index (accounts for both the richness and evenness of species in a sample), observed OTUs (number of detected species), and PD whole tree (reflecting not just the number of species, but also their evolutionary relationships, giving more diversity weight to the samples that include taxa from distinct branches of the tree). * *p* < 0.05.

**Figure 2 nutrients-17-02230-f002:**
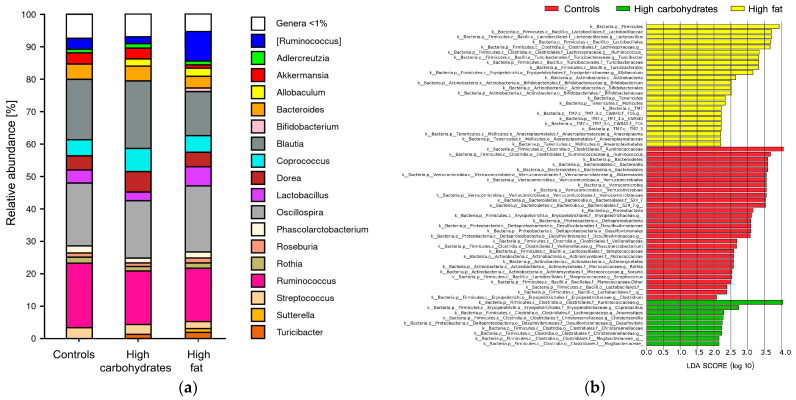
Relative abundance and LDA score of gut bacteria in Sprague Dawley rats maintained on a control or high-fat diet (HFD) for 12 weeks: (**a**) Comparison of abundances of bacterial groups at the genus level; *n* = 44; (**b**) LDA score. A score higher than 2 means differences exist in a given group in terms of Taxonomic rank compared to the other groups; *n* = 44.

**Table 1 nutrients-17-02230-t001:** Nutritive value, crude nutrients, and metabolized energy of diets.

	Control Diet	HDD	HFD
Metabolized energy [kcal/kg]	3514	3772	4497
fat [%]	10	12	45
protein [%]	24	18	18
carbohydrates [%]	66	70	37
moisture [%]	7.9	5.0	3.9
crude ash [%]	4.3	4.2	3.9
crude fiber [%]	3.1	1.5	5.6
crude fat [%]	4.0	5.0	22.6
crude protein [%]	20.7	17.1	20.8
Nitrogen free extractives [%]	60	67.2	43.2

Diets used in this study were obtained from Altromin in Lage, Germany. The complete nutritional value and chemical composition of each diet can be found in Appendix A.

**Table 2 nutrients-17-02230-t002:** Metabolites panel in urine in Sprague Dawley rats maintained on control diet, HDD, or HFD for 12 weeks.

	Controls	HDD	HFD	ANOVA
TMA (µM)	0.5321 ± 0.059	0.6771 # ± 0.067	0.9702 *** ± 0.09	*p* = 0.0004
TMAO (µM)	187.22 ± 14.14	191.57 # ± 15.21	253.41 ** ± 17.59	*p* = 0.006
betaine (ng/mL)	28,566.60 ± 3019.83	33,927.60 ## ± 3347.51	72,751.20 *** ± 10,674.89	*p* < 0.0001
choline (ng/mL)	10,420.80 ± 1339.84	10,535.90 ± 1755.04	10,716.20 ± 1956.70	*p* = 0.94
carnitine (ng/mL)	31,979.62 ± 3031.68	30,233.11 ### ± 3117.92	63,532.91 *** ± 4741.20	*p* < 0.0001
GPC (ng/mL)	13,207.2 ± 2416.60	10,455 ± 1765.87	10,030.6 ± 1825.31	*p* = 0.63

** *p* < 0.01, *** *p* < 0.001 vs. controls; # *p* < 0.05, ## *p* < 0.01, ### *p* < 0.001, HDD vs. HFD; Means ± SE are presented. TMA, trimethylamine; TMAO, Trimethylamine-*N*-oxide; GPC, L-α-Glycerophosphorylcholine; HDD, high-disaccharides diet, HFD, high-fat diet.

**Table 3 nutrients-17-02230-t003:** Daily urinary excretion of metabolites in Sprague Dawley rats maintained on a control diet, HDD, or HFD for 12 weeks.

	Controls	HDD	HFD	ANOVA
TMA (µM)	5.03 ± 0.47	4.06 ## ± 0.28	7.15 * ± 0.86	*p* = 0.0028
TMAO (µM)	1691.00 ± 91.07	1241.54 *## ± 135.63	1801.00 ± 136.85	*p* = 0.0064
betaine (ng/mL)	269,715.00 ± 26,738.18	206,823.00 ### ± 20,958.44	515,994.88 *** ± 68,415.18	*p* < 0.0001
choline (ng/mL)	97,859.03 ± 14,579.89	69,866.68 ± 13,900.51	77,331.77 ± 14,645.93	*p* = 0.36
carnitine (ng/mL)	318,281.70 ± 36,763.11	192,099.90 *### ± 23,948.72	458,937.54 * ± 38,512.07	*p* < 0.0001
GPC (ng/mL)	136,672.30 ± 31,934.2	67,552.85 ± 13,125.88	71,405.41 ± 13,139.23	*p* = 0.08

* *p* < 0.05, *** *p* < 0.001 vs. controls; ## *p* < 0.01, ### *p* < 0.001, HDD vs. HFD; Means ± SE are presented.

**Table 4 nutrients-17-02230-t004:** Metabolites panel in plasma in Sprague Dawley rats maintained on a control, high-disaccharides diet (HDD), or high-fat diet (HFD) for 12 weeks.

	Controls	HDD	HFD	ANOVA
TMA (µM)	0.00639 ± 0.00044	0.00866 ± 0.00144	0.07204 ± 0.04179	*p* = 0.15
TMAO (µM)	1.09 ± 0.10	0.93 ± 0.10	1.01 ± 0.07	*p* = 0.50
betaine (ng/mL)	4512.00 ± 179.17	5992.23 ** ± 280.54	5211.65 ± 358.98	*p* = 0.002
choline (ng/mL)	515.63 ± 35.02	628.64 ± 56.41	575.82 ± 47.60	*p* = 0.26
carnitine (ng/mL)	3365.42 ± 395.09	3706.53 ± 488.86	3141.81 ± 321.34	*p* = 0.60
GPC (ng/mL)	1205.41 ± 84.72	1385.37 ± 136.01	1150.16 ± 141.75	*p* = 0.36

** *p* < 0.01 vs. controls; Means ± SE are presented.

**Table 5 nutrients-17-02230-t005:** Correlation between bacteria and metabolites in HDD and HFD groups.

Taxonomy		HFD				HDD
		Choline	GPC	TMA	TMAO	TMA
Phylum	k__Bacteria;p__Tenericutes			−0.7235 (*p* = 0.0457)		
Class						
Order	k__Bacteria;p__Proteobacteria;c__Gammaproteobacteria;o__Pasteurellales	0.8105 (*p* = 0.0137)	0.7746 (*p* = 0.0420)		0.7746 (*p* = 0.0420)	
	k__Bacteria;p__Actinobacteria;c__Actinobacteria;o__Bifidobacteriales					0.8122 (*p* = 0.0129)
Family	k__Bacteria;p__Proteobacteria;c__Gammaproteobacteria;o__Pasteurellales;f__Pasteurellaceae	0.8105 (*p* = 0.0273)			0.7746 (*p* = 0.0420)	
	k__Bacteria;p__Bacteroidetes;c__Bacteroidia;o__Bacteroidales;f__S24-7				0.7813 (*p* = 0.0420)	
	k__Bacteria;p__Actinobacteria;c__Actinobacteria;o__Bifidobacteriales;f__Bifidobacteriaceae					0.8122 (*p* = 0.0257)
Genus	k__Bacteria;p__Actinobacteria;c__Actinobacteria;o__Bifidobacteriales;f__Bifidobacteriaceae;g__Bifidobacterium					0.8122 (*p* = 0.0484)

Correlations between bacterial taxa and urine metabolite levels in different experimental groups. The table presents the significant Spearman correlations (a correlation coefficient with the corresponding *p*-value).

## Data Availability

The original contributions presented in this study are included in the article/Appendix A. Further inquiries can be directed at the corresponding authors.

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
