# Peer review of "High-Fat Diet with Normal Caloric Intake Elevates TMA and TMAO Production and Reduces Microbial Diversity in Rats"

_nutrients, 2025, doi:10.3390/nu17132230_

Round 1
Reviewer 1 Report
Comments and Suggestions for Authors
nutrients-3700168
The authors evaluated the effect of diets on the production of TMAO and TMA by gut microbiota. The main idea seems interesting, but the form that the authors presented does not allow for a correct evaluation of the merit.
There are problems with redaction and correct use of grammar, and nonsense words and use of acronyms without prior explanation.
Line 30: physiolocial???
The graphical abstract is not explanatory per se also, why did they draw a liver if they did not have any experiment related to this organ?
Line 149: observed_otus???
The authors should explain better Figure 1 and the logic behind this experiment.
In Figure 2B, there are some clusters of bacteria that are repeated in three tree diets group (related to abundance), the authors should inform that and then explain in the discussion.
In “3.5” the entire paragraph must be rewritten (correlation should be explained), and report how the results are prevented from being mere coincidences and that the data have a reciprocal relationship (urine vs. microbiota).
Author Response
Dear Reviewer, thank you for your thorough review and valuable comments that have helped us to improve the manuscript. Please find below answers to your specific comments.
The authors evaluated the effect of diets on the production of TMAO and TMA by gut microbiota. The main idea seems interesting, but the form that the authors presented does not allow for a correct evaluation of the merit.
There are problems with redaction and correct use of grammar, and nonsense words and use of acronyms without prior explanation.
We have carefully reviewed the manuscript and corrected all punctuation and grammatical errors. Additionally, all acronyms have been defined specifically in line 328, in accordance with the journal’s formatting guidelines."
Line 30: physiolocial???
It has been corrected to physiological
The graphical abstract is not explanatory per se also, why did they draw a liver if they did not have any experiment related to this organ?
The graphical abstract has been revised
The graphical abstract, shows the general scheme of metabolism of bacterial metabolites. First TMA is formed in the intestine from its precursors, and then, through the portal vein goes to the liver where it is metabolized to its oxide - TMAO. The manuscript focuses on these two metabolites, and the liver is the production site one of them.
Line 149: observed_otus???
In the corrected manuscript we have explained the abbreviations line 338 , page 12.
"Observed_otus" is a result of alpha diversity analysis that measures community richness without considering abundance. OTUs (Operational Taxonomic Units) stands for operational taxonomical units. In a literal sense, this metric can be interpreted as the number of unique organisms (or species) detected in a given sample/environment. The term "observed_otus" is the default label used by the software.
The authors should explain better Figure 1 and the logic behind this experiment.
Thank you for the feedback. We have modified the description to make the figure clearer.
- Changes in gut microbiota composition in Sprague Dawley rats maintained on control, and high-fat (HFD) diet for 12 weeks: (a) Beta diversity plot showing the shift in the gut microbiota using a weighted UniFrac and an unweighted UniFrac at the start and the end of experiment. Both metrics are used to compute distances between samples and then visualize them using Principal Coordinates Analysis (PCoA). Unweighted UniFrac measures beta diversity based on the presence or absence of taxa, focusing on community membership and giving weight to rare organisms. In contrast, weighted UniFrac incorporates both presence and relative abundance, highlighting differences in community structure and emphasizing dominant taxa. (b) Alfa diversity metrics showing community richness in studied groups: Shannon index (accounts for both the richness and evenness of species in a sample), observed OTUs (number of detected species) and PD whole tree (reflects not just the number of species, but also their evolutionary relationships, giving more diversity weight to samples that include taxa from distinct branches of the tree).
In Figure 2B, there are some clusters of bacteria that are repeated in three tree diets group (related to abundance), the authors should inform that and then explain in the discussion.
In the Figure 2B, all entities are unique. However some of them may look as being the same for example:
k__Bacteria_p__Bacteroidetes_c__Bacteroidia_o__Bacteroidales_f__S24-7
k__Bacteria_p__Bacteroidetes_c__Bacteroidia_o__Bacteroidales_f__S24-7_g_
In this case we have two different taxa. During the Qiime protocol we have applied “open_OTUs_picking” method that clusters sequences by first matching them to a reference database and then de novo clustering any unmatched sequences, allowing both known and novel taxa to be identified. In above example f_S24-7 stand for microorganism that was identified exactly, only on a family level, however f__S24-7_g_ was categorized as unclassified genus within the S24-7 family.Such OTUs_picking method is recommended for non-human microbiomes.
In “3.5” the entire paragraph must be rewritten (correlation should be explained), and report how the results are prevented from being mere coincidences and that the data have a reciprocal relationship (urine vs. microbiota).
To minimize the generation of random associations during the analysis of such a large amount of data and parameters, the following measures were applied. First, the data analysis was performed on normalized data to avoid issues related to the impact of differences in sequencing depth on the quality of comparisons. Additionally, p-values were corrected by the Benjamini-Hochberg FDR procedure for all statistical tests performed in this study.
At the phylum level, only one statistically significant correlation was observed. Tenericutes were negatively correlated with TMA under HFD conditions (r = -0.7235, p = 0.0457).
At the order level, under HFD, Pasteurellales showed a strong positive correlation with choline (r = 0.8105, p = 0.0137), GPC (r = 0.7746, p = 0.0420), and TMAO (r = 0.7746, p = 0.0420). Additionally, under HDD, Bifidobacteriales demonstrated a significant positive correlation with TMA (r = 0.8122, p = 0.0129).
At the family level, Pasteurellaceae, a member of the Pasteurellales order, was positively correlated with GPC (r = 0.8105, p = 0.0273) and TMAO (r = 0.7746, p = 0.0420) under HFD. Furthermore, the S24-7 family within the Bacteroidales order was positively correlated with TMAO (r = 0.7813, p = 0.0420) under HFD. Under HDD, Bifidobacteriaceae exhibited a strong positive correlation with TMA (r = 0.8122, p = 0.0257).
At the genus level, Bifidobacterium, within the Bifidobacteriaceae family, was specifically associated with TMA under HDD conditions, showing a positive correlation (r = 0.8122, p = 0.0484).

Reviewer 2 Report
Comments and Suggestions for Authors
The authors conducted a study of the effects of HFD on TMAO/TMA on rats without obesity. This is an interesting topic. However, the manuscript requires substantial revisions.
The abstract illustration depicts a small part of the study and should be more descriptive of the entire study.
Multiple spelling errors exist.
The methods are not well described. Please explain in detail how "physiologic caloric intake" was conducted and all methods of urine and blood collection. Provide an experimental timeline for when procedures were performed.
Supplementary materials are referenced but I see none.
Describe the types of fat and carbohydrates in the diets.
Table 2: There are significance symbols in the table that are not explained in the footnote.
All abbreviations need definition in table footnotes.
Comments on the Quality of English Language
Spelling errors exist.
Author Response
Reviewer 2
Dear Reviewer, thank you for your thorough review and valuable comments that have helped us to improve the manuscript. Please find below answers to your specific comments.
The authors conducted a study of the effects of HFD on TMAO/TMA on rats without obesity. This is an interesting topic. However, the manuscript requires substantial revisions.
The abstract illustration depicts a small part of the study and should be more descriptive of the entire study.
Multiple spelling errors exist.
We have carefully reviewed the manuscript and corrected all punctuation and grammatical errors
The methods are not well described. Please explain in detail how "physiologic caloric intake" was conducted and all methods of urine and blood collection. Provide an experimental timeline for when procedures were performed.
We have corrected the description as recommended by the Reviewer, methods section have been expanded Line: 75-96,Page: 3
All experimental groups received an equivalent caloric intake, adjusted according to body weight gain as the rats matured. The rats were weighed weekly, and their energy needs were calculated based on their current weights using guidelines from the Nutrient Requirements of Laboratory Animals 1. Each day, they were provided with a specific amount of feed (in grams) corresponding to the calculated caloric requirement. The rats were housed in cages with a fixed amount of food tailored to their body weight. Any uneaten food was collected and weighed daily, with the leftover amount converted into caloric values. The actual energy intake for each rat over the course of the experiment was determined by subtracting the caloric value of the uneaten food from the total calories initially provided.
Rats were initially anesthetized via intraperitoneal (IP) injection of a ketamine-xylazine mixture (70 mg/kg ketamine and 10 mg/kg xylazine). Once fully anesthetized, blood was collected directly from the heart using a syringe.
Blood samples were collected using chilled EDTA tubes and centrifuged at 5000 rpm for 5 minutes at 4 °C. The resulting plasma was transferred into Eppendorf tubes and stored at −20 °C. Following blood collection, the rats were euthanized by decapitation, and stool samples (0.5 mL) were obtained from the midsection of the colon. The animals were fasted for 6 h before sacrifice.
Additionally : According to Nutrient Requirements of Laboratory Animals1 We have assumed the energy demand 200 kcal EN /BWkg0,75/day about 266 kcal EM (1-4 week) and 180 kcal EN /BWkg0,75/day about 240 kcal EM (week 4-12), (net energy units, can be converted to ME units by dividing by 0.75, based on an efficiency of 75 percent in conversion of ME to net energy).
- National Research Council (US) Subcommittee on Laboratory Animal Nutrition. Nutrient Requirements of Laboratory Animals: Fourth Revised Edition, 1995(National Academies Press (US), 1995). https://doi.org/10.17226/4758.
Supplementary materials are referenced but I see none.
We are very sorry that some data from the supplement was not uploaded successfully during the submission of the manuscript in the system. The relevant data can be found now in supplementary materials.
Supplementary table 1. Average total energy and food intake.
|
|
Average total energy intake /rat/experiment (kcal) |
Average total food intake/rat/experiment (g) |
|
Control diet |
4560.06 |
1297.68 |
|
High-disaccharide diet |
4520.90 |
1198.54 |
|
High-fat diet |
4368.76 |
971.48 |
Describe the types of fat and carbohydrates in the diets.
The exact compositions were as follows:
|
Carbohydrates |
Control diet |
High-disaccharide diet |
High-fat diet |
|
Monosaccharides |
15.143 mg/kg |
66.500 mg/kg |
102.200mg/kg |
|
Disaccharides |
117.705 mg/kg |
441.105 mg/kg |
50.355 mg/kg |
|
Polysaccharides |
427.227 mg/kg |
133.527 mg/kg |
229.252mg/kg |
|
Fatty acid |
Control diet |
High-disaccharide diet |
High-fat diet |
|
Arachidic acid C-20:0 |
340 mg/kg |
50 mg/kg |
0 |
|
Eicosanoic acid C-20:1 |
153 mg/kg |
150 mg/kg |
0 |
|
Alpha-Linolenic acid C-18:3 |
333 mg/kg |
150 mg/kg |
1.582 mg/kg |
|
Linolenic acid C-18:2 |
2.059 mg/kg |
28.500 mg/kg |
2.260 mg/kg |
|
Palmitic acid C-16:0 |
5.240 mg/kg |
2.500 mg/kg |
5.017 mg/kg |
|
Stearic acid C-18:0 |
3.793 mg/kg |
1.350 mg/kg |
13.786 mg/kg |
|
Oleic acid C-18:1 |
11.300 mg/kg |
13.500 mg/kg |
38.872 mg/kg |
Full details of the diets can be found in the supplementary materials.
Table 2: There are significance symbols in the table that are not explained in the footnote.
All abbreviations need definition in table footnotes.
All abbreviations are given at the end of the manuscript, according to a schedule prepared by the publisher. In addition, in the table description we have expanded all abbreviations in accordance with the reviewer's recommendations.
TMA, trimethylamine; TMAO, Trimethylamine-N-oxide; GPC, L-α-Glycerophosphorylcholine; HDD, high-dissacarides diet, HFD high-fat diet.

Reviewer 3 Report
Comments and Suggestions for Authors
- Prior studies (e.g., Yoo et al., Science, 2021) have already shown gut-derived TMAO elevation from fat-rich diets, including mechanisms involving colonocyte dysfunction. This study fails to significantly extend these insights.
- The manuscript strongly implies that TMAO elevations are linked to Pasteurellaceae and S24-7 taxa but does not demonstrate expression of TMA-producing genes (e.g., cutC) in these organisms.
- Correlations do not equal causation, and without metagenomics, transcriptomics, or functional microbial validation (e.g., qPCR of cutC, cntA), these claims remain speculative.
- The authors emphasize that plasma TMAO levels did not change despite major urinary excretion differences, which contradicts the primary conclusion that HFD increases systemic TMAO burden.
- The relevance of urinary TMAO as a proxy for cardiovascular risk is questionable without corroborating plasma or tissue-level data.
Author Response
Dear Reviewer, thank you for your thorough review and valuable comments that have helped us to improve the manuscript. Please find below answers to your specific comments.
- Prior studies (e.g., Yoo et al., Science, 2021) have already shown gut-derived TMAO elevation from fat-rich diets, including mechanisms involving colonocyte dysfunction. This study fails to significantly extend these insights.
Reviewer gives as an example a very good paper1 showing a mechanism how a high-fat diet escalates Escherichia coli choline catabolism by altering intestinal epithelial physiology.
Our study differs in various aspects, and adds new data to existing knowledge. Specifically, in contrast to the study cited by the Reviewer:
In our experiment we used rats not mice and it is well recognized that TMA metabolism differed significantly between mice and rats2-6.
- Yoo, W., Zieba, J. K., Foegeding, N. J., Torres, T. P., Shelton, C. D., Shealy, N. G., Byndloss, A. J., Cevallos, S. A., Gertz, E., Tiffany, C. R., Thomas, J. D., Litvak, Y., Nguyen, H., Olsan, E. E., Bennett, B. J., Rathmell, J. C., Major, A. S., Bäumler, A. J., & Byndloss, M. X. (2021). High-fat diet-induced colonocyte dysfunction escalates microbiota-derived trimethylamine N-oxide. Science (New York, N.Y.), 373(6556), 813–818. https://doi.org/10.1126/science.aba3683
- Maksymiuk KM, Szudzik M, Samborowska E, Chabowski D, Konop M, Ufnal M. Mice, rats, and guinea pigs differ in FMOs expression and tissue concentration of TMAO, a gut bacteria-derived biomarker of cardiovascular and metabolic diseases. PLoS One. 2024 Jan 24;19(1):e0297474. doi: 10.1371/journal.pone.0297474. PMID: 38266015; PMCID: PMC10807837.
- Lattard V, Lachuer J, Buronfosse T, Garnier F, Benoit E. Physiological factors affecting the expression of FMO1 and FMO3 in the rat liver and kidney. Biochem Pharmacol. 2002 Apr 15;63(8):1453-64. doi: 10.1016/s0006-2952(02)00886-9. PMID: 11996886.
- Veeravalli S, Karu K, Scott F, Fennema D, Phillips IR, Shephard EA. Effect of Flavin-Containing Monooxygenase Genotype, Mouse Strain, and Gender on Trimethylamine N-oxide Production, Plasma Cholesterol Concentration, and an Index of Atherosclerosis. Drug Metab Dispos. 2018 Jan;46(1):20-25. doi: 10.1124/dmd.117.077636. Epub 2017 Oct 25. PMID: 29070510; PMCID: PMC5733448
- Cherrington NJ, Cao Y, Cherrington JW, Rose RL, Hodgson E. Physiological factors affecting protein expression of flavin-containing monooxygenases 1, 3 and 5. Xenobiotica. 1998 Jul;28(7):673-82. doi: 10.1080/004982598239254. PMID: 9711811.
- Janmohamed A, Hernandez D, Phillips IR, Shephard EA. Cell-, tissue-, sex- and developmental stage-specific expression of mouse flavin-containing monooxygenases (Fmos). Biochem Pharmacol. 2004 Jul 1;68(1):73-83. doi: 10.1016/j.bcp.2004.02.036. PMID: 15183119.
The aim of this study was to evaluate the effect of high-fat and high-disaccharide diets change gut microbiome, but without animals developing obesity as it is stated in introduction line 53-57, page 2. This was possible as we controlled total calorie intake to match physiological needs. Finally, in this study we have also included a high-carbohydrate diet.
- The manuscript strongly implies that TMAO elevations are linked to Pasteurellaceae and S24-7 taxa but does not demonstrate expression of TMA-producing genes (e.g., cutC) in these organisms.
We agree with Reviewer that the presence of the taxa Pasteurellaceae and S24-7 is not sufficient evidence for their direct involvement in TMAO production.1 In the corrected manuscript this has been stressed in the limitation section:
Although we observed correlations between urine TMA/TMAO levels and the rela-tive abundance of Pasteurellaceae and S24-7, it is important to note that taxonomic associations alone do not confirm functional involvement in TMA production.
While these findings emphasize the influence of diet on microbiota composition and TMA/TMAO production, there is currently no conclusive evidence in literature that Pas-teurellaceae, S24-7, Tenericutes, or Bifidobacteriales directly produce TMA from precur-sors such as choline or L-carnitine. Enzymes essential for this process, such as cutC, have not been confirmed in their genomes. Further metagenomic or transcriptomic analyses are needed to elucidate the potential role of these microbial groups in TMA/TMAO metabolism
- Jameson, E., Quareshy, M., & Chen, Y. (2018). Methodological considerations for the identification of choline and carnitine-degrading bacteria in the gut. Methods (San Diego, Calif.), 149, 42–48. https://doi.org/10.1016/j.ymeth.2018.03.012
- Correlations do not equal causation, and without metagenomics, transcriptomics, or functional microbial validation (e.g., qPCR of cutC, cntA), these claims remain speculative.
We agree with the reviewer that, despite our best efforts to ensure the accuracy of the correlation analyses, we recognize that correlation does not imply causation.
We included these statement in the Study limiations.
To minimize the generation of random associations during the analysis of such a large amount of data and parameters, recommended measures were applied. First, the data analysis was performed on normalized data to avoid issues related to the impact of differences in sequencing depth on the quality of comparisons. Additionally, the analysis took into account p-values corrected by the Benjamini-Hochberg FDR procedure for all statistical tests performed in this study. This correction is commonly used to control for multiple hypothesis testing errors when doing statistical tests on microbiome data.
- The authors emphasize that plasma TMAO levels did not change despite major urinary excretion differences, which contradicts the primary conclusion that HFD increases systemic TMAO burden.
We appreciate the reviewer’s comment but respectfully disagree with the conclusion. The stable plasma TMAO levels likely reflect efficient renal excretion rather than contradicting the observed increase in systemic production. It is well-established that plasma concentrations of metabolites are tightly regulated when renal function remains intact. In this context, the elevated urinary TMAO excretion observed in the HFD group is a clear indicator of increased systemic production and turnover. Therefore, 24-hour urinary excretion is a more reliable marker of total TMAO burden than plasma levels alone. A useful parallel can be drawn with high-protein diets: in healthy individuals, such diets result in markedly increased production of ammonia and urea, yet plasma levels remain within normal ranges due to effective renal clearance. However, in the setting of kidney dysfunction, accumulation of these metabolites becomes evident. Similarly, significant elevations in plasma TMAO are typically observed only when renal excretory function is impaired, such as in aging or renal disease, incuding cardiovascular disease leading to renal mulfunciton.
The relevance of urinary TMAO as a proxy for cardiovascular risk is questionable without corroborating plasma or tissue-level data.
As pointed out above, elevated plasma TMAO is often a marker of more advanced cardiovascular disease, frequently accompanied by renal impairment. In contrast, we propose that urinary TMAO may serve as a more sensitive and earlier indicator of increased systemic TMAO production and cardiovascular risk in individuals with preserved renal function. By capturing enhanced metabolic turnover before plasma accumulation occurs, urinary TMA/TMAO offers potential utility in early risk assessment.

Round 2
Reviewer 1 Report
Comments and Suggestions for Authors The authors evaluated the effect of diets on the production of TMAO and TMA by the gut microbiota. Following a request for revision, they made the suggested changes, making the text more fluid and easier to understand. The main idea seems interesting and therefore scientifically publishable.Reviewer 2 Report
Comments and Suggestions for Authors
The authors have made substantial improvements to the manuscript.
A few remaining comments:
Methods: Please specify that the rats were individually housed, if so.
Line 93: "The rats were maintained for 2 days in metabolic cages to evaluate their 24-h fluid and energy balance prior to and after the main section of the study." It is unclear how you evaluated their fluid and energy balance. What exactly was analyzed in feces and urine? Where the rats fasted?
Tables 2 & 3: The significance symbols quadruple hashtag and triple asterisk are not defined.
Table 5 footnotes are incorrect
Please show/describe the beginning and ending body weight of the groups.
Lines 159, 199 contain spelling errors.
Ref 11 is not complete.
Minor spelling errors exist.
Author Response
Dear Reviewer we sincerely appreciate the time and effort you devoted to reviewing our manuscript. Your insightful feedback has been very helpful in improving our work.
A few remaining comments:
Methods: Please specify that the rats were individually housed, if so.
Line 63: The rats were housed in groups of 4.
Line 93: "The rats were maintained for 2 days in metabolic cages to evaluate their 24-h fluid and energy balance prior to and after the main section of the study." It is unclear how you evaluated their fluid and energy balance. What exactly was analyzed in feces and urine? Where the rats fasted?
line 92:
The rats were fasted in the cages before sacrifice, just after the metabolic cages were completed.
Metabolic cages are used to collect, urine, feces and control the consumption of food and water. the rat is placed there alone for two days with water and chow. Then we can calculate the fluid and energy balance for each single animal. In feces from the midsection of the colon we performed the s16 rRNA sequencing. In the urine of rats we analyzed the bacterial metabolites with Waters Acquity Ultra Performance Liquid Chromatograph coupled with a Waters TQ-S triple-quadrupole mass spectrometer.
Tables 2 & 3: The significance symbols quadruple hashtag and triple asterisk are not defined.
It has been done
Table 5 footnotes are incorrect
It has been revised
Line 253: Correlations between bacterial taxa and urine metabolite levels in different experimental groups. The table presents significant Spearman correlations (correlation coefficient with corresponding p-value)
Please show/describe the beginning and ending body weight of the groups.
|
Initial body weight (g) |
185.75 ± 7.34
|
182.26 ± 6.29 |
182.30 ± 5.69 |
P=0.83 |
|
Final body weight (g) |
391.07 ± 26.07
|
384.59 ± 25.62
|
368.40 ± 23.02
|
P=0.67 |
We have also included the table as Supplementary Table 2 and cited it appropriately in the manuscript. Line 86.
Lines 159, 199 contain spelling errors.
It has been corrected
Ref 11 is not complete.
Reference no. 11 has been corrected

Reviewer 3 Report
Comments and Suggestions for Authors
No further Comments.
Author Response
Dear Reviewer,
We would like to sincerely thank you for your thoughtful and thorough review. We deeply appreciate the time and effort you dedicated to evaluating our work. Your comments and suggestions reflect a high level of scientific expertise, and they have been extremely valuable to us in improving the quality of our manuscript.
Best Regards,
Mateusz Szudzik